


# Understanding the Primary Emissions and Secondary Formation of Gaseous Organic Acids in the Oil Sands Region of Alberta, Canada

John Liggio[1], Samar G. Moussa[1], Jeremy Wentzell[1], Andrea Darlington[1], Peter Liu[1], Amy Leithead[1], Katherine Hayden[1], Jason O'Brien[1], Richard L. Mittermeier[1], Ralf Staebler[1], Mengistu Wolde[2] and Shao-Meng Li[1]

[1]Air Quality Research Division, Environment and Climate Change Canada, Toronto, Ontario, Canada, M3H 5T4.
[2]National Research Council Canada, Flight Research Laboratory, Ottawa, Canada K1A 0R6

*Correspondence to*: John Liggio (John.Liggio@Canada.ca)

**Abstract.** Organic acids are known to be emitted from combustion processes and are key photochemical products of biogenic and anthropogenic precursors. Despite their multiple environmental impacts, such as on acid deposition and human/ecosystem health, little is known regarding their emission magnitudes or detailed chemical formation mechanisms. In the current work, airborne measurements of 18 gas-phase low molecular weight organic acids were made in the summer of 2013 over the oil sands region of Alberta, Canada, an area of intense unconventional oil extraction. The data from these measurements were used in conjunction with emission retrieval algorithms to derive the total and speciated primary organic acid emission rates, as well as secondary formation rates downwind of oil sands operations. The results of the analysis indicate that approximately 12 tonnes day$^{-1}$ of low molecular weight organic acids, dominated by $C_1 - C_5$ acids, were emitted directly from off-road diesel vehicles within open pit mines. Although there are no specific reporting requirements for primary organic acids, the measured emissions were similar in magnitude to primary oxygenated hydrocarbon emissions measured previously ($\approx$20 tonnes day$^{-1}$), for which there are reporting thresholds. Conversely, photochemical production of gaseous organic acids significantly exceeded the primary sources, with formation rates up to $\approx$184 tonnes day$^{-1}$ downwind of the oil sands facilities. The formation and evolution of organic acids from a Lagrangian flight were modelled with a box model, incorporating a detailed hydrocarbon reaction mechanism extracted from the Master Chemical Mechanism (v 3.3). Despite evidence of significant secondary organic acid formation, the explicit chemical box model largely underestimated their formation in the oil sands plumes, accounting for 39%, 46%, 26% and 23% of the measured formic, acetic, acrylic and propionic acids respectively, and with little contributions from biogenic VOC precursors. The model results, together with an examination of the carbon mass balance between the organic acids formed and the primary VOCs emitted from oils sands operations, suggest the existence of significant missing oil sands related secondary sources/precursor emissions and/or an incomplete mechanistic and quantitative understanding of how they are processed in the atmosphere.





## 1. Introduction

Organic acids are important atmospheric constituents with various sources, sinks and roles in gas phase, particle phase, and precipitation chemistry. Organic acids are known photochemical products of volatile organic compound (VOC) oxidation, some of which are of significantly low volatility to contribute to secondary organic aerosol (SOA). Numerous studies have identified and/or quantified specific organic acids or organic acid functional groups in particulate matter (PM) (Kawamura and Bikkina, 2016;Zhang et al., 2016;Ho et al., 2015;Li et al., 1997), with organic acids often accounting for a significant fraction of PM mass (Duarte et al., 2015;Yatavelli et al., 2015;Sorooshian et al., 2010). While PM associated organic acids have a long history of measurements (Li and Winchester, 1992), low molecular weight organic acids (LMWOA; $\approx C_1$-$<C_{10}$), found primarily in the gas-phase, have only recently received attention, likely due to difficulties associated with their measurement at trace levels. The advent of real-time instrumentation with organic acid selectivity (Le Breton et al., 2012;Veres et al., 2008;Yatavelli et al., 2012;Lee et al., 2014), and LMWOA satellite retrievals (Cady-Pereira et al., 2014;Shephard et al., 2015;Stavrakou et al., 2012) has helped to shed light on the importance of low molecular weight organic acids in various atmospheric chemistry processes.

From the atmospheric chemical process perspective, LMWOA can be important contributors to precipitation acidity and ionic balance, particularly in remote areas (Khare et al., 1999;Stavrakou et al., 2012), and are expected to become increasingly important as anthropogenic $NO_x$ and $SO_x$ emissions continue to decrease. They are key participants in the aqueous phase chemistry of clouds and contribute to secondary organic aerosol (SOA) formation through various reactions within the aqueous portion of the particle-phase (Carlton et al., 2007;Ervens et al., 2004;Lim et al., 2010). Furthermore, since organic acids can be photochemical products, they can also serve as indicators of atmospheric transformation processes. Real-time and improved measurements for LMWOA present a means to test the validity of current photochemical reaction mechanisms of volatile organic compounds (VOC) from which the LMWOA are ultimately derived. From an environmental and health perspective, deposition of LMWOA may have ecosystem impacts as they have been shown to be toxic to various marine invertebrates (Staples et al., 2000;Sverdrup et al., 2001), phytotoxic (Himanen et al., 2012;Lynch, 1977), and interfere with the uptake and mobilization of heavy metals by microbial communities in soils (Song et al., 2016;Menezes-Blackburn et al., 2016). However, studies on the human toxicity of LMWOAs are sparse and the results unclear (Rydzynski, 1997;Azuma et al., 2016).

The sources of gaseous organic acids are highly varied. Primary sources of organic acids (LMWOA in particular) include gasoline and diesel vehicle exhaust emissions (Kawamura et al., 2000;Zervas et al., 2001a;Wentzell et al., 2013;Crisp et al., 2014;Zervas et al., 2001b), biofuel combustion, biological activity (ie: direct vegetation emissions), soil emissions, and biomass burning (Chebbi and Carlier, 1996). However, the largest source of LMWOAs is likely to be their formation from the oxidation of VOCs (Paulot et al., 2011;Veres et al., 2011). The most important oxidation pathway leading to their formation is likely to be the gas phase ozonolysis of alkenes, where the further reaction of the resulting stabilized biradicals with water (the dominant pathway under tropospheric conditions) will form organic acids as a primary product (Neeb et al., 1997). Other mechanistic routes include the





OH initiated oxidation of aromatic species (Praplan et al., 2014), the decomposition of PAN (Surratt et al., 2009), and the OH oxidation of vinyl alcohols (Andrews et al., 2012). Some volatile organic acids have also been shown to be evolved to the gas-phase through heterogeneous aerosol reactions (Molina et al., 2004) and/or aqueous chemical reactions involving glyoxal (Carlton et al., 2007).

Despite the known primary and secondary sources of LMWOA, the chemistry forming LMWOA from their precursors remains poorly understood. Incorporation of known emissions and associated chemistry into global models has repeatedly indicated missing sources, particularly with respect to formic and acetic acids (FA and AA) (Ito et al., 2007;Paulot et al., 2011;von Kuhlmann et al., 2003). While it has been stated that secondary sources of FA and AA from biogenic precursors (isoprene in particular) dominate the global budget of these species, studies

have demonstrated that large inconsistencies exist between measurements and model predictions in the northern hemisphere (Paulot et al., 2011). Potential inconsistencies have not been examined for other LMWOA. However, given the limited measurements of other LMWOAs, as well as the complex and likely incomplete mechanistic understanding of their formation, larger discrepancies for other LMWOAs are expected.  Large model-measurement inconsistencies in the northern hemisphere, after having included biogenic secondary sources, suggest the possibility

of secondary formation from anthropogenic VOC precursors accounting for an increased fraction of the global LMWOA budget. This points to the need for further measurements of LMWOA formed downwind of anthropogenic emissions.

         One such anthropogenic source of LMWOA precursors is the oil and gas sector, which has been expanding significantly in North America over the last several decades, and has significant VOC precursor emissions (Gilman et al., 2013). Indeed, the formation of a single LMWOA (FA) in an oil and gas region of the United States has been

studied recently (Yuan et al., 2015) during the winter season. The results of that study demonstrated that FA is significantly under-predicted when using the current FA photochemical mechanism, even after accounting for potential heterogeneous and aqueous formation. Despite an entirely different VOC profile compared to urban areas, secondary FA formation remained high in the region, suggesting that anthropogenic VOC precursors from the oil

and gas sector are important contributors to FA formation.

         The oil sands (OS) region of Alberta, Canada represents another important oil and gas producing region in North America with relatively large VOC emissions(Li et al., 2017). It is estimated to contain up to 1.7 trillion barrels of highly viscous oil mixed with sand (Government of Alberta, 2009) from which oil is recovered through unconventional surface mining or in-situ steam assisted extraction.  Rising OS oil production has raised concerns

over its environmental impacts, including those associated with the deposition of toxic compounds, and the acidification of nearby ecosystems via the deposition of $SO_x$ and $NO_x$ (Jung et al., 2013;Kelly et al., 2009;Kirk et al., 2014).  Recent evidence has also indicated that the downwind transformation of OS gaseous precursors (intermediate volatility compounds in particular; IVOC) to SOA represents a large PM input into the atmosphere (Liggio et al., 2016).  The same photochemical processes which give rise to the observed SOA will also lead to

various other gas-phase photochemical products that include organic acids.  Given that the OS activities are





surrounded by forests, the OS represents an ideal location to study the primary emission and secondary formation of gaseous organic acids from both industrial and biogenic precursors, in the absence of other confounding emissions.

In this work, we describe airborne measurements of gaseous LMWOA both directly emitted from OS activities (ie: primary) and formed via secondary reactions of various OS emitted and/or biogenic hydrocarbon

precursors. Through the use of specifically tailored flight patterns and top-down emissions retrieval algorithms, the total (and speciated) primary emission and secondary production rates of LMWOA from the OS are derived; from which the relative importance of primary and secondary organic acids to the total organic acid budget in OS plumes are evaluated. Through Lagrangian box modelling of successive OS plume intercepts, the relative importance of biogenic and anthropogenic precursors within OS plumes is examined, providing a means to evaluate our current

understanding of organic acid photochemical formation mechanisms for selected LMWOA. It is expected that the results of this study will be broadly applicable to the secondary formation of LMWOA from other anthropogenic activities with hydrocarbon emissions.

## 2.0 Methods

### 2.1 Aircraft Campaign

Airborne measurements of a variety of air pollutants aboard the National Research Council of Canada (NRC) Convair-580 aircraft were performed over the Athabasca oil sands region of northern Alberta from August 13 to September 7, 2013 in support of the Joint Canada-Alberta Implementation Plan on Oil Sands Monitoring (JOSM). Details regarding the overarching study objectives, aircraft campaign implementation and technical aspects

have been described previously (Gordon et al., 2015;Liggio et al., 2016). During this study, 22 flights were conducted over the oil sands for a total of approximately 84 hours. Thirteen of the flights were conducted to quantify facility overall primary emissions by flying a box of four or five sided polygons, at multiple altitudes, resulting in 21 separate virtual boxes around 7 oil sands facilities. In addition, 3 flights (F7, F19, F20) were conducted to study the photo-chemical transformation of pollutants downwind of the OS (See supporting

information), including the secondary formation of organic acids.

### 2.2 Gaseous Organic Acid Measurements

Gaseous organic and inorganic acid measurements were conducted aboard the aircraft with a High Resolution Time-of-Flight Chemical Ionization Mass Spectrometer (HR-ToF-CIMS; Aerodyne Research Inc.). A

detailed description of the instrument and principles of operation has been given elsewhere (Bertram et al., 2011;Lee et al., 2014) and in the supporting information. Briefly, the HR-ToF-CIMS used in this study was a differentially pumped Time-of-Flight mass spectrometer configured to use acetate ion as the reagent ion in the ionization of molecules of interest (Veres et al., 2008;Brophy and Farmer, 2015) via the following reaction



$$CH_3COO^- + HA \rightarrow CH_3COOH + A^- \qquad (1)$$

where HA is the acid of interest and $A^-$ is the respective anion. Thus, acids with a gas-phase acidity greater than that of acetic acid (eq 1) will be ionized and subsequently sent into the mass spectrometer for detection. There is also evidence that acetate dimers can be involved in the ionization process and/or a number of clustering-

declustering/deprotonation reactions of acetate with organic acids (Brophy and Farmer, 2015). At the high field strengths (ie: strong declustering conditions) utilized during this study (Yuan et al., 2016;Brophy and Farmer, 2015), it is expected that such reactions ultimately also lead to a deprotonated acid ion ($A^-$ in Eq 1). Calibrations of organic acids were conducted both in the field and post study using a liquid calibration unit (LCU; Ionimed Analytic), which provided a stable gas stream of analytes by volatilizing aqueous organic acid standards of known composition and

concentration prepared from pure compounds (>99%; Sigma Aldrich). A list of the quantified organic acids is given in Table 1, and includes 18 acids spanning the $C_1 - C_{10}$ range.

The HR-ToF-CIMS data were processed using the TofWare software program (Tofwerk AG, Switzerland), using an approach for mass calibration and high resolution peak fitting which has been described previously (Brophy and Farmer, 2015;Yuan et al., 2016), resulting in an overall conservative uncertainty of ~40% in the

quantified species (Yuan et al., 2016). The accuracy of the mass ($m/z$) calibration was approximately $5 - 7$ ppm during the campaign. The reagent acetate ion signal during this campaign was approximately $0.9–2.5 \times 10^6$ counts per second (cps), and was used to normalize the analyte signals during post processing. The mass resolution of the HR-ToF-CIMS during the study was approximately 3000 to 4000 for ions spanning m/z=100 to greater than $m/z$=200. The quantified ions were predominantly those whose signal dominated the nominal m/z space, minimally perturbed

by neighboring shoulders and thus less affected by uncertainties associated with HR peak fitting (Yuan et al., 2016). Despite the high mass resolution, isomeric species (ie: the same exact mass) which are also acidic in nature (ie: ionisable) cannot be resolved from each other. For some organic acid species this is not relevant, as no other acidic compounds with the same exact mass are likely to exist in the atmosphere. Such species include formic acid, propionic acid, acrylic acid and pyruvic acid. For all other species, the signal at the exact mass was calibrated with

the available surrogate standards listed in Table 1. While this introduces a degree of uncertainty, calibrated response factors across all species and m/z generally did not vary by more than a factor 2-3, and it is expected that organic acids with the same exact mass will have response factors which vary considerably less than this.

A Proton Transfer–Time Of Flight–Mass Spectrometer (PTR-ToF-MS, Ionicon AnalytiK) was used to measure acetic acid (which cannot be measured with the HR-ToF-CIMS) in real-time during each flight. The PTR-

ToF-MS is a soft ionization technique that detects VOCs with a proton affinity greater than that of water. This includes species such as unsaturated hydrocarbons, aromatics, and various oxygenated compounds. Details of the PTR-TOF-MS technique and application, have been described previously (Graus et al., 2010;De Gouw and Warneke, 2007;Li et al., 2017). The deployment of the PTR-ToF-MS on the aircraft during this study, its calibration, and data processing have been described elsewhere (Li et al., 2017). Acetic acid via the PTR-ToF-MS

was calibrated using a method described previously (De Gouw et al., 2003;Zhao and Zhang, 2004), by calculating a response relative to a known and calibrated reference compound (toluene in this case). The response factor for acetic acid ($R_{AA}$) is calculated as:



$$R_{AA} = \frac{k_{AA} T_{AA} R_{Tol}}{k_{Tol} T_{Tol}} \qquad\qquad (2)$$

Where $k$ refers to the kinetic rate constant for the reaction of acetic acid or toluene with $H_3O^+$ , T is the experimentally determined ion transmission efficiencies ($C_2H_5O_2^+$; $m/z = 61$ and $C_7H_9^+$; $m/z = 93$) and $R_{Tol}$ is the experimentally derived response factor for toluene (using a standard gas cylinder). The response factor for acetic acid calculated in this manner was within 10% of that derived using a permeation device during previous studies.

**2.3. Other Supporting Measurements.**

A detailed description of the meteorological variables, aircraft state parameters and full list of gas/particle measurements is provided elsewhere (Gordon et al., 2015;Liggio et al., 2016) (http://jointoilsandsmonitoring.ca/default.asp?lang=En&n=A743E65D-1). The current work makes use of a subset of these measurements. These include refractory black carbon (rBC, also referred to as BC) measurements via a Single Particle Soot Photometer (SP2; Droplet Measurement Technologies, Boulder, CO, USA.), VOC canister sampling followed by off-line analysis and nitrogen oxides (NO and $NO_2$) and ozone measurements (TECO 42i-TLand 49i respectively; Thermo Fisher Scientific, Waltham, MA, USA). Details with respect to supporting measurements are provided in the supporting information.

**2.4 Topdown Emission Rate Retrieval Algorithm (TERRA).**

An algorithm (TERRA) designed to estimate pollutant transfer rates through virtual boxes/screens from aircraft measurements was used to derive the primary and secondary emission/production rates from the oil sands flights (Gordon et al., 2015 and SI). Briefly, primary emissions are derived with utilizing box-like aircraft flight patterns surrounding each of the main surface mining facilities, pollutant measurements at high time resolution, and wind speed, direction, temperature, and pressure data. Organic acid measurements, at high time resolution, can be used directly in TERRA for estimating their primary emissions; however they are also secondary products which may form via chemistry in the short distance/time between the source area and the virtual box wall. Hence, direct use of TERRA for estimating primary organic acid emissions may result in their overestimation. Alternatively, organic acid primary emissions are estimated after normalization to BC in the source area and scaling by the BC emissions via an approach which is described in section 3.1.1.

Secondary formation rates of organic acids were also derived with an extended version of TERRA using Lagrangian transformation flights (F19, F20). The extended TERRA quantifies the mass transfer rate of pollutants (kg/hr) across the virtual screens of transformation flights, in the same manner as for virtual box type flights, and has been described in detail for secondary organic aerosol (SOA) (Liggio et al., 2016) and in the supporting information. Briefly, for the transformation flights, TERRA is applied to single screens created by stacking horizontal legs of flight tracks at multiple altitudes and spirals. Concentration data are mapped to the screens and interpolated using a simple kriging function. Pollutant concentrations are also extrapolated below the lowest flight altitude (150 m agl)





based on the assumption of a well-mixed layer below the lowest flight track. It has been demonstrated that this extrapolation is the main source of uncertainty in TERRA, resulting in overall uncertainty in the derived pollutant transfer rates of approximately 20% (Gordon et al., 2015;Liggio et al., 2016).

**2.5 Box Modelling**

A photochemical box model (AtChem-Online; atchem.leeds.ac.uk) coupled with the Master Chemical Mechanism (MCM v 3.3; University of Leeds, http://mcm.leeds.ac.uk/MCM/) (Jenkin et al., 2012) was used to simulate the individual organic acid (and total) formation and evolution during transformation flight 19, as well as ozone and free radical production, and other photochemical products. The specific data points within the plume to simulate with the model were selected to be those which were very close to being truly Lagrangian in nature as determined from the wind speed, wind direction and flying time. The successive plume intercepts modelled here are depicted in Figure S-1, where the same air parcel was sampled typically within 1 minute of its arrival time based on the wind. The model consisted of an explicit mechanism for 4578 species in 18045 reactions. Further details with respect to modifications to the MCM for organic acids are given in the supporting information and in Yuan et al., 2015. Photochemical rate constants were calculated using Tropospheric Ultraviolet and Visible Radiation Model (TUV; https://www2.acom.ucar.edu/modeling/tropospheric-ultraviolet-and-visible-tuv-radiation-model) and were constrained along the flight path. Relative humidity, temperature and pressure were also constrained with measurement data along the flight path. CO and biogenic VOCs (isoprene, α- pinene and β- pinene) were constrained via measurements along the entire flight path of the aircraft to account for biogenic emissions between screens. Measured cycloalkanes were lumped into cyclohexane (the only cycloalkane in MCM v3.3), and any other VOC not present in the MCM was lumped into a VOC with similar reactivity in the MCM. The model was initialized and constrained using the measurements of VOCs (including organic acids), NOx, CO, $O_3$, $SO_2$ and other parameters at the first screen. The model was run for 3 hours to correspond with the flight time, and with a time step of 1 minute. Black carbon measurements were used to derive first order dilution rate constants, and applied to all species to account for ongoing dilution. Species within the plume were diluted at every time step with air outside the plume whenever background measurements of the acids were available, otherwise clean air was used as background.

**3.0 Results and Discussion.**

**3.1.0 Primary LMWOA Emission Sources**

Primary emissions of various pollutants from specific surface mining OS facilities were estimated by flying virtual boxes around operations followed by subsequent analysis using TERRA (Gordon et al., 2015;Liggio et al., 2016;Li et al., 2017). The specific OS facilities which were evaluated are shown in Table S-1, with corresponding flight numbers, and include Syncrude, Suncor, CNRL, Shell and Imperial Oil. The geographical locations of these



operations in relation to the entire OS region are provided elsewhere (Li et al., 2017). The results of a typical emission flight (F18; Table S-1) are shown in Figure 1 for a $C_5$ organic acid, after having applied the Simple Kriging method to derive a continuous concentration surface. The enhancement in the concentration of this $C_5$ acid on the downwind box wall (Figure 1) clearly demonstrates that this species is associated with OS activities from this

particular facility. Similar enhancements for all other measured organic acids at this facility (and others) and during other flights of Table S-1 were also observed. While this organic acid and most others are clearly derived from an OS source, generally, surface mining operations consist of a variety of potential pollutant sources as outlined in Figure 1, including open pit mines, tailing ponds and processing plants. These sources are all often in close proximity to each other. However, the chemical nature, location and extent of individual OS plumes within facilities,

relative to the prevailing winds can often be used to determine specific sources. For example, while mines and tailings ponds will lead primarily to surface emissions, processing plant emissions may be mostly from elevated stacks, particularly for $SO_2$ (Gordon et al., 2015). Furthermore, the majority of black carbon emissions within OS operations arise from the heavy hauler trucks used in the open pit mining (Wang et al., 2016) and significantly less from other combustion sources within the processing plants. In the case of the LMWOA measured during emission

box flights, the prevailing winds and positioning of the concentration enhancements at the box wall of each flight indicate that the open pit mines are the most likely source of the LMWOA emissions (for example in Figure 1). In addition, organic acids in box flights were consistently correlated with black carbon (BC), and not correlated with other species such as $SO_2$. An example of this relationship is given in Figure 2-A for propionic acid during F18. Figure 2-A demonstrates that there is a high degree of similarity between the time series of propionic acid and BC,

particularly at the plume intercepts on the virtual box wall. This is consistent with BC being a major emission from the large diesel heavy hauler trucks within the mine (Wang et al., 2016), and the known emissions of various LMWOA from diesel fuel combustion (Kawamura et al., 2000;Wentzell et al., 2013;Crisp et al., 2014;Zervas et al., 2001a). This relationship is also observed for other emission flights, and for all other LMWOAs measured. The correlation between LMWOA and BC is subsequently used to derive individual and total LMWOA primary

emission rates from various OS facilities (after accounting for potential secondary formation), as described further below.

### 3.1.1 Primary LMWOA Emission Rate Estimates

For conserved and chemically unreactive species, the virtual box flights can be used directly in TERRA to estimate emissions (kg/hr) via a mass balance approach. However, the distance between OS sources and any given

box wall in an emission flight can range from 10 to 15 km. With the average wind speeds during these flights, such a distance corresponds to approximately 10 – 60 minutes in transport time. During this time, it is possible that photochemical reactions occur which could increase or decrease the concentration of a given pollutant at the box wall, and hence affect the final emission rate calculated by TERRA. This has been accounted for in the case of VOCs by estimating the rates of oxidation for specific hydrocarbons during travel to the exit of the virtual box,

using known rate constants for the reaction of VOC with OH and $O_3$ (ie: $k_{OH}$ and $K_{O3}$) (Li et al., 2017). This has been shown in most instances to result in small corrections to the TERRA derived emission rates for the hydrocarbons (Li





et al., 2017).  The degradation of organic acids during transport to the box wall is expected to be slow since their OH and $O_3$ rate constants are generally small. However, they are more importantly, photochemical products of VOC oxidation such that attempting to correct for the contribution of their photochemical formation prior to input into TERRA is not feasible as it requires detailed knowledge of all oxidation mechanisms leading to LMWOA and their

associated yields. Such information is not available and would nonetheless carry a high degree of uncertainty.

Alternatively, BC is used as a normalizing tracer to estimate facility emission rates. As noted above, the time series of LMWOA and BC are similar (Eg:Fig 2-A), suggesting a common incomplete combustion source from the large mining trucks. The correlation between propionic acid and BC for F18 (for example) is shown in Figure 2-B, and indicates that while the two species are correlated, a significant spread in the data exists (denoted by the $25^{th}$

– $75^{th}$ percentiles), likely caused by a degree of photochemical propionic acid formation while BC is conserved. This is also reflected in the observation that other LMWOAs are more significantly correlated with each other (Figure 2-C) than they are with BC (ie: LMWOA are all more similarly formed and lost relative to BC).  These observations further imply that the formation of LMWOAs will be reflected in the evolution of the background subtracted LMWOA to BC ratio ($\Delta$LMWOA/$\Delta$BC) between the emission source and virtual box wall. In this case, the

$\Delta$LMWOA/$\Delta$BC at the source (ie: the emission ratio) can be used to estimate facility primary LMWOA emissions ($E_{LMWOA}$; kg/hr), when the corresponding BC emissions $E_{BC}$ from the virtual boxes over the individual facilities have been determined using TERRA (Cheng et al., 2017) , according to Eq 3:

$$E_{LMWOA} = \left(\frac{\Delta LMWOA}{\Delta BC}\right)_{Source} \times E_{BC} \qquad (3)$$

LMWOA emission ratios (($\Delta$LMWOA/$\Delta$BC)$_{source}$) were derived from flights where horizontal transects between the

center of the open pit mines and the exiting box walls were flown.  These flights included F17, F18 and F21 (spanning 4 facilities) and are depicted in Figures 3, S-2 and S-3.  These figures indicate the horizontal extent of the BC plumes from the various mines, and the approximate times and distances from the approximate centre of a given mine to the various horizontal transects. At the closest transects to the mine centers, the time from emission was estimated to range from approximately 0.3 – 6 minutes based on the average wind speeds during these flights; a time

during which photochemical production of the LMWOA is expected to be minimal and the ($\Delta$LMWOA/$\Delta$BC) ratio should approach the true emission ratio, unlike that expected during the longer transport time from emission to the box wall (10 – 60 min; see Li et al. (2016)) The evolution of $\Delta$LMWOA/$\Delta$BC for the 4 depicted OS operations, at the various horizontal transects of Figures 3, S-2 and S-3 (eg: A, B, C, $A_1$, $B_1$, $C_1$ ) are shown in Figure 4 for a selected organic acid. The boxes in these figures represent the $25^{th}$ to $75^{th}$ percentiles of the individual

$\Delta$LMWOA/$\Delta$BC ratio values within the plume only, as defined spatially by the BC (which is approximately zero outside of the plume), and at approximately the same altitude. These figures demonstrate a degree of photochemical LMWOA formation when moving from the emission sources to the virtual box walls. Hence, the emission ratios for the various species are considered to be the ratio in the yellow highlighted regions of Figure 4 (ie: the closest approach to the mine source - A, $A_1$ and $A_2$).



Photochemical formation of organic acids from co-emitted hydrocarbons within the mines is not expected to significantly contribute to the derived emissions ratios in the 0.3 to 6 min travel time. This is particularly likely for the transects closest to the mine sources, as very high co-emitted NO at the source titrates $O_3$ to levels below 15 ppbv (Figure S-4) and is likely to effectively supresses active photochemistry at the source via OH radical.

Furthermore, most LMWOA are likely later generation products of VOC oxidation and not likely formed to a great extent in such a short time. Nonetheless, a small secondary LMWOA contribution from co-emitted hydrocarbons to the emission ratios here cannot be entirely ruled out.

The emission ratios ($\Delta$LMWOA/$\Delta$BC) derived for individual LMWOA species for the 4 facilities shown in Figure 4 are presented in Figure S-5, where the error bars represent the 25th to 75th percentiles of the computed ratio

data. Generally, the profiles of emission ratios in Figure S-5 are relatively similar to each other, regardless of the facility (means within 15 – 70 % for individual species). However, the differences between facilities may be attributed to differing emissions control systems between recent and older model mining vehicles. The largest emission ratios are associated with formic and acetic acids, ranging from 412 to 800 pptv/$\mu$gm$^{-3}$, followed by propionic, butyric and pentanoic acids in the range of 114 to 205 pptv/$\mu$gm$^{-3}$. Other LMWOAs have smaller but

non-negligible emission ratios. The corresponding facility LMWOA emission rates (kg hr$^{-1}$) derived with Eq (3) (a primary emission rate estimate) are shown in Figure S-6. Where individual emission ratios for a given facility were not available, the mean of the emission ratios for each species is used to compute the emission rate (ie: for Suncor – MS and Imperial – KL). Accordingly, the largest primary emissions rates are observed for formic and acetic acids (65 and 129 kg hr$^{-1}$; summed across facilities), followed by propionic, butyric and pentanoic acids at 52, 54 and 40

20     kg hr$^{-1}$ respectively. This corresponds to the fractional contributions to the total measured LMWOA mass emission rate shown in Figure S-7A. where formic and acetic acids alone account for ~43% of the primary emissions.

From a facility standpoint, speciated and total measured primary emissions of LMWOA during this study were significantly variable (Figure S-6), with total LMWOA emissions for the Suncor-MS, Syncrude-ML, Syncrude-AU, Shell-MR/JP, CNRL-HOR and Imperial-KL facilities estimated be 162±22, 108±15, 45±6, 56±8,

60±8  and 19±3 kg hr$^{-1}$ respectively. The total primary LMWOA emission rates should also then be somewhat proportional to OS oil production, given that LMWOA are associated with diesel exhaust during mining activities. Since BC emission rates were found to be linearly correlated with the quantity of oil sands mined (Cheng et al., 2017), LMWOA emission rates derived using BC emission rates as the tracer are therefore similarly correlated with the oil sands mined. This correlation suggests that primary LMWOA emissions may also be expected to track

increases or decreases in OS productivity.

The sum of the measured emission rates for LMWOA (ie: total primary) is shown in Figure 5, as well as the measured primary emissions of total and oxygenated VOCs (Li et al., 2017). In total, the OS are estimated to emit approximately 12 tonnes day$^{-1}$ of primary LMWOAs (see supporting information). Relative to the total VOC emitted simultaneously ($\approx$214 tonnes day$^{-1}$), the emission of LMWOA is small, representing an increase to the total

accounted for VOC emissions of less than 6%. However, of the total VOC emissions, oxygenated VOCs have been


estimated to account for <10% (≈20 tonnes day$^{-1}$) (Li et al., 2017) (Figure 5), and comprised mainly of methanol, formaldehyde and acetone. Hence, the LMWOA emissions here represent an increase of up to 60% to the oxygenated VOC mass emissions, which have previously been unaccounted for, and for which there are currently no regulatory reporting requirements.

The primary LMWOA emissions from the OS are not easily placed in context, as measurements of emission rates from various other anthropogenic sources are generally not available. On a global scale, primary sources of LMWOAs have been estimated to be very small relative to their photochemical production from a variety of precursor gases (Paulot et al., 2011). This indicates that while possibly important on a regional scale, the primary emitted LMWOAs associated with the OS are not expected to contribute significantly to the overall organic acid

atmospheric burden both in the Canadian context and/or globally. Given the large emissions of VOCs from OS operations (Li et al., 2017), it is likely that secondary formation of LMWOAs from OS derived precursors is more significant than their primary emissions, a subject to be further explored below.

### 3.2.0 Secondary formation of LMWOA in OS plumes

### 3.2.1 Secondary Formation rates

        The secondary formation rates of LMWOAs are estimated using a modified version of TERRA with transformation flights 19 and 20 (see methods, section 2.5). An example of the resultant screens from which secondary formation rates are derived is given in Figure 6 for $C_5H_8O_2$. Using BC measurements to define the OS mine plume spatial dimensions (red boxes; Figure 6), the formation rates for any specific LMWOA is given as the

difference in transfer rates between screens (1 to 4). Accordingly, for $C_5H_8O_2$, approximately 219±43 kg hr$^{-1}$ is formed downwind of the OS, in the 3 hours between screens 1 and 4. Similarly, the formation rates for all measured LMWOAs between screens 1 and 4 during F19 are shown in Figure 7A, along with the sum of the estimated primary LMWOA emissions of Figure S-6 (summed over all facilities). From this Figure it is evident that primary emissions of LMWOAs are negligible compared to that formed via oxidation of precursors; the sum of primary emission rates

(across all facilities) are on average 30 times lower than the transfer rates through screen 4 (Figure 7A). Consequently, the total speciated secondary formation rates (between the source area and screen 4; ie: ≈4 hrs) are given as the transfer rates at screen 4 (kg hr$^{-1}$), after having subtracted the small primary emission contributions; these are shown as the green bars of Figure 7A. During this 4 hr mid-day time period, individual secondary production rates of LMWOAs ranged from ≈20 – 6700 kg hr$^{-1}$, dominated by acetic (≈6700 kg hr$^{-1}$) and formic acids

(≈3200 kg hr$^{-1}$), with all other species in the ≈20 - 500 kg hr$^{-1}$ range. On a percent basis, formic and acetic acids together account for ~70% of the total estimated secondary LMWOAs production rate downwind of the oil sands, with all other organic acid species generally contributing less than 3% each to the total (Figure S-7B). This is in contrast to the estimated primary emission rates from the OS, where formic and acetic acids accounted for ~43% of the total measured primary LMWOA emission rate, with several other species accounting for up to 12% (Figure S-

7A). Such relative differences between LMWOA secondary production and primary emission rate profiles is





expected, since photochemical formation of LMWOA is likely to occur at rates, and via formation mechanisms that will differ from that of primary combustion sources. Furthermore, formic and acetic acids being the lowest MW acids are likely to have many more precursors than other LMWOAs, implying that their relative proportions to the total LMWOA should increase with increasing photochemical processing.

The hourly formation rates of Figure 7A are further scaled to one photochemical day (ie: tonnes day$^{-1}$) as described in detail previously (Liggio et al., 2016). While oil Sands operations are active 24 hours per day, a simple scaling up by multiplying the hourly rates by 24 may add significant uncertainty, as photochemistry does not occur uniformly across these hours.  Alternatively, the hourly LMWOA production rate (for each hour) was scaled by the time integrated OH radical concentration, which was modelled with a box model (Liggio et al., 2016). The daily

LMWOA production rate is then the sum of the scaled hourly production rates, and is shown in Figure 7B for flights 19 and 20. Typically, the scaled daily secondary production rates were ~54% of the daily rate derived using a simple 24 hour multiplier; 184±37 tonnes day$^{-1}$ and 173±34 tonnes day$^{-1}$ for F19 and F20 respectively. This estimation neglects the dry deposition of LMWOA which may be significant. Accounting for the potential deposition of these species (see Supporting information) increases the estimated total secondary LMWOA formation rates even further

to 288±58 and 226±45 tonnes day$^{-1}$ respectively. The total LMWOA secondary production rate (excluding deposition, as a lower limit) is also compared to the total measured hydrocarbon VOC emissions from OS operations in Figure 7B. Although, on a total mass basis (tonnes day$^{-1}$) LMWOA production is almost as large as the total VOC primary emissions (~214 tonnes day$^{-1}$)(Li et al., 2017) at least 2 oxygen atoms are added during oxidation (to form acids), and hence a carbon mass comparison is more relevant and is also shown in Figure 7B.

This comparison indicates that up to 50% of the organic carbon mass emitted is transformed into organic acids (ie: an effective yield of approximately 50%) in one photochemical day. Conversely, typical yields for LMWOAs from VOCs in smog chamber experiments are less than 10-15% and often much lower (Paulot et al., 2009;Neeb et al., 1997;Butkovskaya et al., 2006;Berndt and Böge, 2001;Yuan et al., 2015). This suggests that the total primary VOC emissions have been significantly underestimated and that numerous other hydrocarbon species

have not been measured, despite having quantified >150 individual VOC species. This is consistent with the pool of semi- and intermediate volatility compounds (SVOC/IVOC) expected to be emitted (but not measured) which were responsible for the observed SOA in OS plumes (Liggio et al., 2016). The ability of semi- and intermediate volatility hydrocarbons to form LMWOAs is unknown. However, IVOC/total VOC emission ratios for the OS have been suggested to be large (Liggio et al., 2016), indicating that yields of LMWOAs from these species can have a

significant impact on the total LMWOA production rate.  However, from a carbon mass balance perspective, even LMWOA yields of ≈10% from IVOCs would require that the total emitted carbon in the form of IVOCs exceeds the measured VOC carbon emissions (≈175 tonnes C day$^{-1}$; Figure 7B) by at least a factor of 5 in order to produce the ≈75 tonnes C day$^{-1}$ of LMWOA estimated in Figure 7B. While there is no evidence to the contrary, such a large IVOC contribution to total OS emissions may be unlikely.  This may suggest LMWOA yields from IVOCs may be

larger than typical VOCs. In this regard, recent evidence has suggested that molecular fragmentation during the oxidation of IVOCs specifically, can dominate over functionalization (and hence SOA formation) in significantly



less than a photochemical day (Lambe et al., 2012). Molecular fragmentation to smaller species such as LMWOAs in the highly photo-chemically active plumes encountered here (Liggio et al., 2016), could significantly increase LMWOA yields relative to laboratory yields from experiments which are typically performed at moderate OH and for less than a photochemical day (Paulot et al., 2009;Praplan et al., 2014).  Biogenic hydrocarbons present along the flight track in F19 and F20 also have the potential to contribute to the observed LMWOA production rate, and have not been included in the estimates of total OS VOC emissions (Figure 7B)(Li et al., 2017). However, largangian box modelling of F19 indicates that the biogenic contribution to several LMWOAs is relatively small as described in Section 3.2.3.

The current LMWOA secondary production rates represent the first such estimates downwind of a large scale industrial facility. Indeed, measurements of LMWOA secondary production rates from any source are generally not available, and hence placing the LMWOA production from OS precursors in context with that of other sources is difficult. Available estimates of secondary anthropogenic production rates of LMWOAs have been limited to formic and acetic acids, and only in a global context (Ito et al., 2007;Paulot et al., 2011). Nonetheless, the most recent estimates indicate that the sum of secondary anthropogenic and biomass burning sources of FA and AA contribute approximately 6.3 and 1.3 MegaTonnes per year (Mtonnes yr$^{-1}$) respectively to the global budget (Paulot et al., 2011). Crudely downscaling (365 days/yr), results in a global daily anthropogenic (+ biomass burning) production rate of approximately of $2.1 \times 10^4$ tonnes day$^{-1}$ (sum of FA and AA). Hence, the combined FA and AA secondary production rates observed in OS plumes (~129 tonnes day$^{-1}$) likely contribute <1% to the global secondary anthropogenic/biomass burning budget. While only qualitative, this comparison suggests that the OS are not a major anthropogenic secondary source of FA and AA globally.

The impact of the OS on a smaller scale (e.g.: regional), as a photochemical producer of FA and AA (and other organic acids) is not clear since comparative data are not available. However, we note that the total LMWOA formed within a photochemical day of the OS in this study (up to ≈184 tonnes day$^{-1}$ or ≈288  tonnes day$^{-1}$ if accounting for deposition) is comparable to the total SO$_2$ previously reported to be emitted from the OS (~200-300 tonnes day$^{-1}$)(Hazewinkel et al., 2008;Jung et al., 2011).  The strong acidity associated with sulfur deposition is likely to dominate downwind ecosystem effects. Consequently, critical load exceedances for highly sensitive aquatic systems in Northern Alberta have mainly been assessed from the perspective of sulfur and nitrogen (Cathcart et al., 2016;Whitfield et al., 2016). The impact of the large amount of weak organic acidity formed downwind of the OS has not been evaluated, and may have a relevant impact on ecosystem acidification in highly sensitive systems approaching their respective sulfur and/or nitrogen critical loads. These results warrant a further investigation of the potential impact of this LMWOA emission/formation in this context.

### 3.2.3 Modelling secondary LMWOA formation in OS plumes

While organic acid secondary production rates are large during F19 and F20 relative to OS hydrocarbon primary emissions (Fig 7), the specific precursor species leading to these observed acids were not clear, including





the impact of BVOC oxidation. However, the Lagrangian nature of F19 allows a comprehensive box model evaluation of the most recent photo-chemical mechanisms leading to LMWOAs and an estimate of precursor contributions. Flight 19 was modelled for 3 hours, beginning at the first screen and ending at the fourth screen, utilizing measurements of all VOCs and inorganic gases at screen 1 as the initial conditions. The most recent Master

Chemical Mechanism V3.3, with further improvements for FA and AA mechanisms, was used to simulate the secondary chemistry. A detailed description of the box modelling approach is given in methods (section 2.5). As demonstrated in Figure S-8, the box model effectively simulates the evolution of some known organic and inorganic gases, including the oxidative loss of alkenes, alkanes and aromatics (and others), the cycling of NOx and ozone formation; this provides confidence in the interpretation of other aspects of the model output. The comparison

between measured and modelled LMWOAs during F19 is shown in Figure 8. The model output is compared with four specific organic acids (formic, acetic, acrylic and propionic) as these species are certain to be free of isomeric organic acid interferences in the HR-ToF-CIMS measurements. Figure 8 indicates that the formation of these four species is poorly simulated by the box model. The model/measurement time series diverge quickly (<30 min), with the model output dominated by dilution (causing the downward trend) while the observed acids increase over 3

15    hours, suggesting that the formation rate of these species is sufficient to overcome the effects of dilution and deposition. After 3 hours, the model under-predicts the concentrations of formic, acetic, acrylic and propionic acids in the plume by factors of 2.6, 2.2, 3.9 and 4.4 respectively. Similarly, the total measured LMWOAs are also under-predicted (Figure 8) by a factor of 2.9, where 27 modelled organic acids account for 99.8% of the LMWOA mass as compared to the 18 measured species. Accounting for the depositional losses for these species in the model will

increase the model-measurement discrepancy even further. Such poor model-measurement agreement is again consistent with significant unaccounted for VOC/IVOC emissions from the OS which lead to various LMWOAs as was suggested by the unrealistically high effective organic acid yield of 50% in Figure 7B (Section 3.2.1). This is particularly true for FA, whose formation in the MCM has been recently updated (Yuan et al., 2015) and yet remains poorly simulated.

The model output also allows for a more detailed examination of the precursors responsible for a portion of the organic acids observed in OS plumes. The relative contribution of various precursor types to the modelled formic, acetic, acrylic and propionic acids after 3 hours during F19 (screen 4) is shown in Figure 9. These contributions by the precursors indicate that the oxidation of biogenic emissions along the flight track were largely not major contributors to the observed levels of these species relative to the oxidation of OS emissions. Formic and

acetic acids have received significant attention recently with respect to their photochemical production mechanisms from biogenic species, and have been updated in the current MCM used here (see methods). Despite this update, the combined oxidation of isoprene and monoterpenes accounted for approximately 18% and 33% of the formic and acetic acid produced after 3 hours, with measured aromatics, alkenes and alkanes accounting for ~2 − 11% each. The biogenic contributions to acrylic and propionic acids are even smaller (Figure 9); with isoprene oxidation

contributing ~3% to acrylic acid formation and propionic acid having no biogenic precursor contribution. The completeness/validity of the MCM with respect to acrylic and propionic acids is unclear; it has not been recently updated, as few experimental studies exist to validate MCM yields of these species from various precursors.





However, yields from biogenic VOCs for these two species are not likely to exceed those of FA and AA, making the contribution to acrylic and propionic acids small.

Undoubtedly, the largest contribution to these four LMWOAs is unaccounted for and labelled as "missing" in Figure 9. Such missing precursor sources account for ~ 54 – 77% of production of these species after 3 hours of processing. Although other processes such as aqueous and heterogeneous chemistry can lead to LMWOAs and are not included in the MCM mechanism here, their contribution to formic acid formation was estimated to be <5% (Yuan et al., 2015). Similarly, fog events and air-snow exchange have no contribution here (ie: fog and snow were not present). For formic acid, this missing contribution is similar in magnitude to that observed in another oil and gas producing region (Yuan et al., 2015). The source of the missing contributions can be a result of several factors; incorrect yields of LMWOA from species currently in the MCM, unknown mechanistic pathways leading to LMWOAs from various MCM precursors and unmeasured hydrocarbon precursors which hence were not included in the MCM. The first factor appears to be minor, and in the case of FA and AA the updated yields from various hydrocarbons in the MCM are considered upper limits (Yuan et al., 2015) and further updates to yield data will likely not result in significantly more FA and AA contributions. For the second factor, LMWOA yields from alkanes are not included in the current MCM (with the exception of cyclic alkanes), although limited evidence suggests some LMWOAs may be indirectly formed via their oxidation (Zhang et al., 2014), but insufficient to account for a significant portion of the missing mass in Figure 9. Given the unrealistically high LMWOA effective yields estimated here (up to 50%) and the small contribution of BVOC oxidation in these plumes, the third factor is the most likely reason for the missing contributors; they may be entirely due to the oxidation of OS related emissions which were not measured. Furthermore, given the known presence of a large amount of lower volatility OS hydrocarbons in these plumes that give rise to SOA (Liggio et al., 2016), it is possible that the large fraction of the missing term will be IVOC in nature, whose photo-chemical mechanisms have yet to be elucidated, but as noted above are susceptible to significant fragmentation (Lambe et al., 2012) potentially leading to LMWOA.

**4.0 Conclusions**

Measurements of 18 gas-phase organic acids have been made in the OS region of Alberta, Canada for the first time, indicating that they are emitted from primary sources and formed via secondary chemistry in OS plumes. The main primary source of these acids is demonstrated to be combustion within open pit mines leading to primary LMWOA emissions of up to 12 tonnes day$^{-1}$ suggesting that primary oxygenated emissions from OS activities are likely larger than previously considered, with organic acids comprising a large fraction of these emissions.

Despite the potential important contribution of primary LMWOA emissions to the OS emissions inventory, secondary formation rates in OS plumes were found to be dominant over primary emission rates by more than an order of magnitude; secondary formation rates within 1 photo-chemical day of the OS were in excess of 180 tonnes day$^{-1}$, and were the first such estimate from any industrial or urban/suburban source. Consequently, up to 50% of the carbon emitted was transformed to organic acids within 1 photo-chemical day; an unusually high effective yield



which suggests the presence unknown/unmeasured hydrocarbons that are capable of producing LMWOAs upon oxidation with significant yields. The Lagrangian nature of several flights during this study also provided a unique opportunity to examine the ability of current photo-chemical mechanisms to reproduce LMWOA observations. Subsequent box modelling of the evolution of several organic acids over the course of 3 hours significantly under-

5 predicts the concentration of simple organic acids. A 'missing' source accounts for the majority of the species measured (54-77%), but is not related to the oxidation of biogenic species. The poor model predictive ability and the lack of biogenic contribution suggests missing OS precursors which are likely to be of intermediate volatility. These results suggest that further work is required to understand the nature of these missing precursors, to elucidate their photo-chemical pathways leading to LMWOA and other products, and to narrow model-measurement gap. Finally,

the impact of the weak acid deposition to sensitive ecosystems and contribution to overall critical load exceedances is not clear, but likely warrants further investigation.

**Acknowledgments.**

We thank the National Research Council (NRC) of Canada flight crew of the Convair-580, the technical support staff of AQRD and Dr. Stewart Cober for the management of the study. The project was supported by the Climate Change and Air Quality Program (CCAP), and the Joint Oil Sands Monitoring program (JOSM).

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





**Table 1.** Molecular formulas and associated species names for organic acids detected during aircraft measurements.

| Molecular Formula | Detected ion : nominal *m/z* | Species name[a] |
|---|---|---|
| HCOOH | HCOO- : 45 | Formic acid |
| $CH_3COOH$ | CH3COO- : 61 | Acetic acid[b] |
| $C_3H_4O_2$ | $C_3H_3O_2^-$ : 71 | Acrylic acid |
| $C_3H_4O_3$ | $C_3H_3O_3^-$ : 87 | pyruvic acid |
| $C_3H_6O_2$ | $C_3H_5O_2^-$ : 73 | Propanoic acid |
| $C_4H_6O_2$ | $C_4H_5O_2^-$ : 85 | methacrylic acid |
| $C_4H_6O_3$ | $C_4H_5O_3^-$ : 87 | 2-Oxo-butanoic acid |
| $C_4H_8O_2$ | $C_4H_7O_2^-$ : 101 | butyric acid |
| $C_5H_6O_3$ | $C_5H_5O_3^-$ : 117 | 4-Oxo-2-pentenoic acid |
| $C_5H_8O_2$ | $C_5H_7O_2^-$ : 113 | 4-pentenoic acid |
| $C_5H_8O_3$ | $C_5H_7O_3^-$ : 99 | levulinic acid |
| $C_5H_{10}O_2$ | $C_5H_9O_2^-$ : 115 | Pentanoic acid |
| $C_5H_{10}O_3$ | $C_5H_9O_3^-$ : 101 | 2-hydroxy-3-methylbutyric acid |
| $C_6H_{10}O_2$ | $C_6H_9O_2^-$ : 113 | cyclopentanoic acid |
| $C_6H_{12}O_2$ | $C_6H_{11}O_{2-}$ : 115 | hexanoic acid |
| $C_7H_{10}O_2$ | $C_7H_9O_2^-$ : 125 | 1-Cyclohexenecarboxylic Acid |
| $C_7H_{12}O_2$ | $C_7H_{11}O_2^-$ :127 | 6-heptenoic acid |
| $C_{10}H_{16}O_3$ | $C_{10}H_{15}O_3^-$ : 183 | Pinonic acid |

**a.** Species name represents the molecule used for calibration, although not all molecular formulas are for unique
species. **b.** Measured by PTR-ToF-MS; see methods.





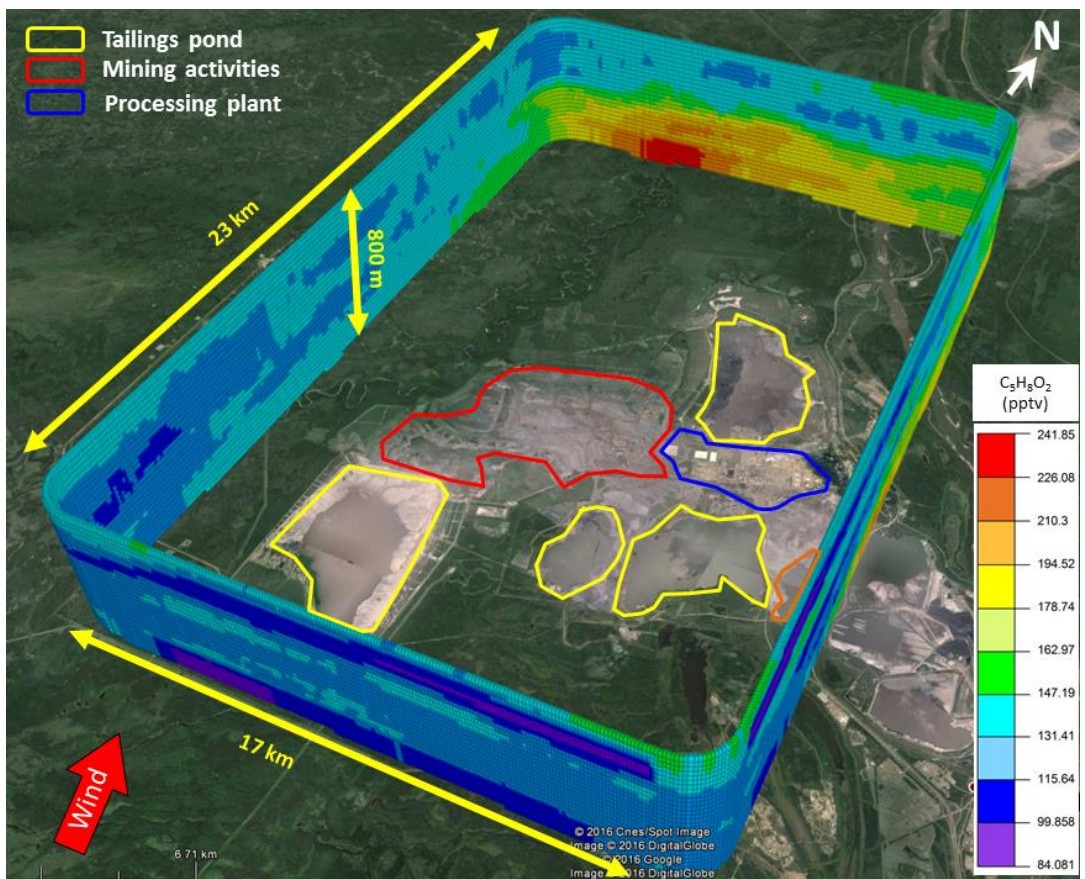

**Figure 1.** Emissions box flight during flight 18 on September 3$^{rd}$, 2013 (Syncrude –ML) as outputted from TERRA for a typical measured organic acid (C$_5$H$_8$O$_2$). Data have been krigged to produce a continuous concentration surface around various OS operation sources.





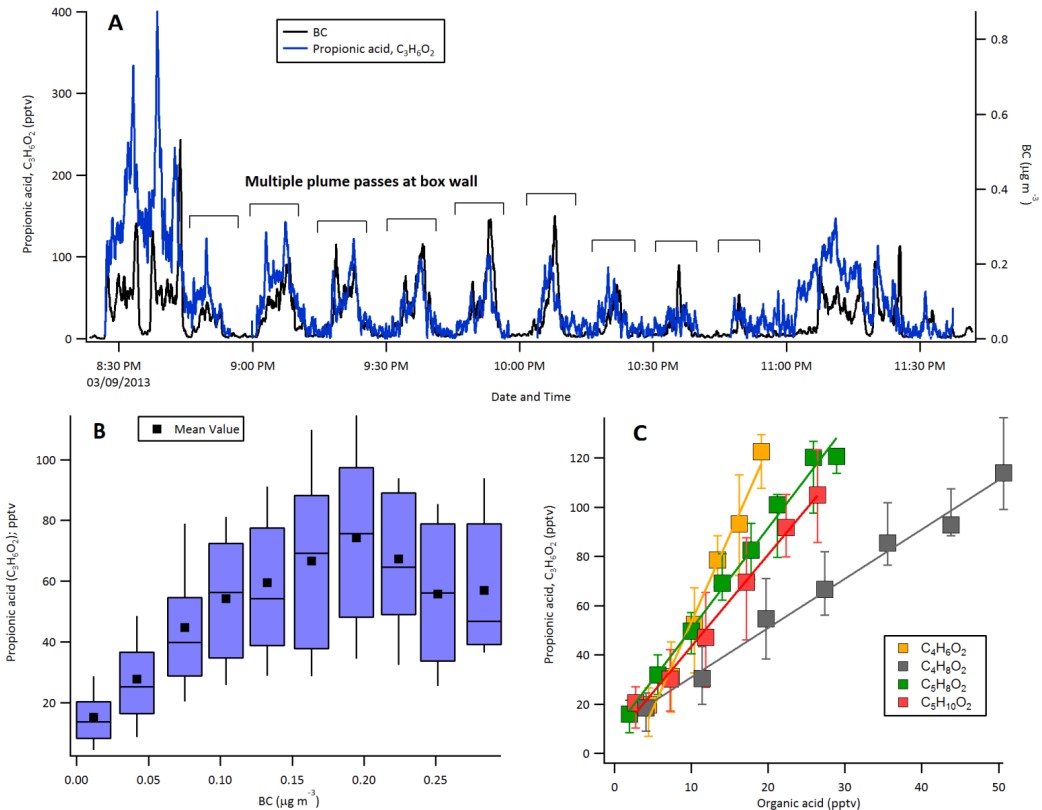

**Figure 2. A.** Time series of measured propionic acid and BC during Flight 18 (Syncrude – ML). Multiple plume intercepts at the virtual box wall are shown. **B.** Correlation between propionic acid and BC during F18. Black data points represent the means while blue boxes and whiskers are the 25[th] to 75[th] percentiles and 90[th] and 10[th] percentiles respectively. **C.** Correlations between various measured organic acids during F18. Error bars represent the 25[th] to 75[th] percentiles of the data.




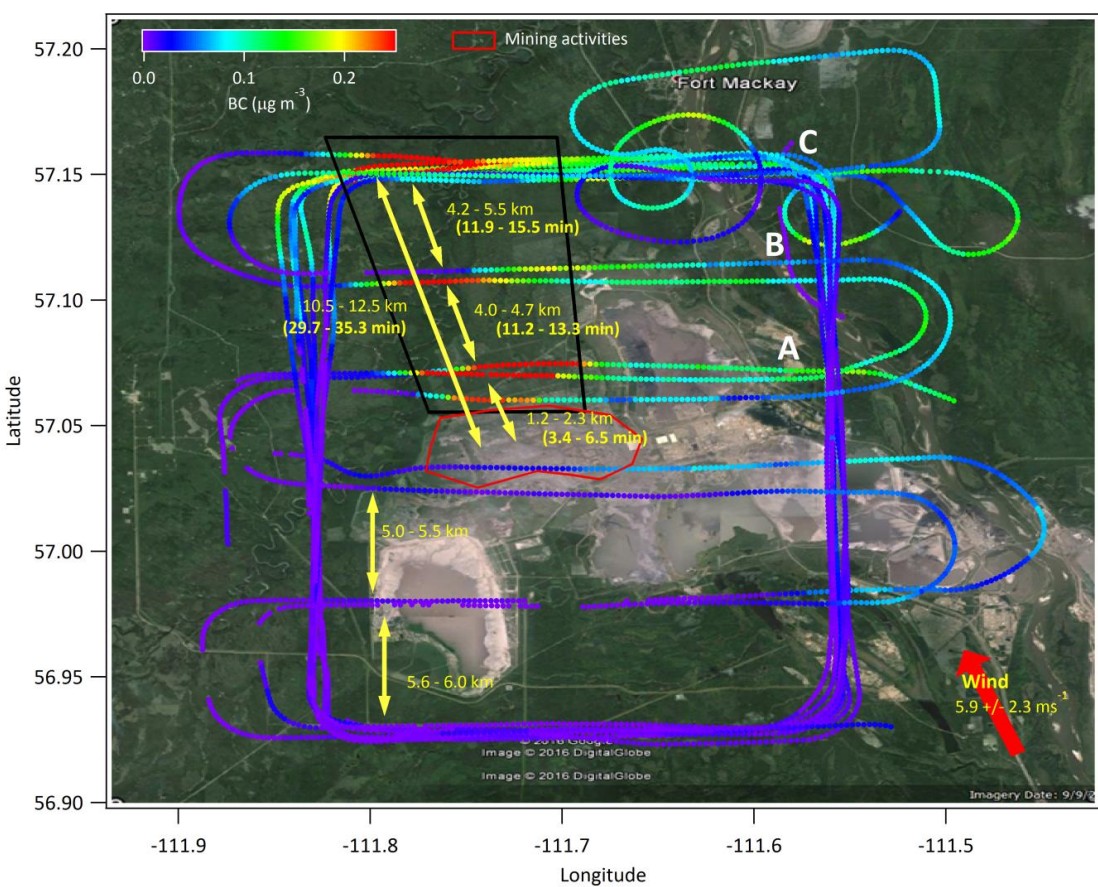

**Figure 3.** Concentration of BC during emission flight 18 (Syncrude – ML), showing horizontal transects (A – C) within the box and closest to the mining emission source (red box). Time in brackets represents the approximate time between horizontal transects or from the approximate center of the mine to the closest transect. The time is calculated based upon the average wind speed during this portion of the flight. The range in values is based upon differences in time calculated from the northern and southern most legs of transects A – C.



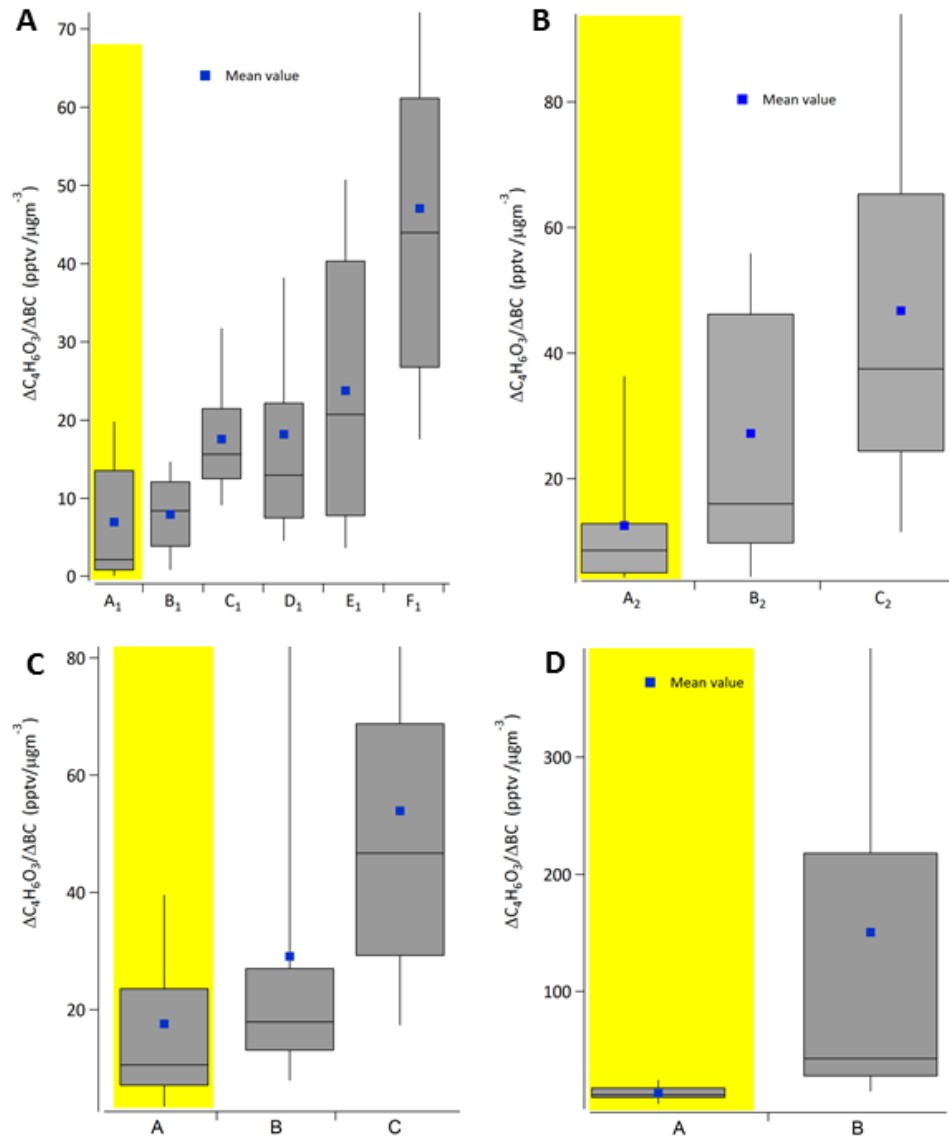

**Figure 4:** $\Delta$LMWOA/$\Delta$BC evolution (using $C_4H_6O_3$ as an example) across the various transects (A, B, C etc…) of emission box flights. Yellow highlighted area represents the emission ratio (ie: closest transect to the open pit mine). Box and whiskers represent 25th, 75th, 90th and 10th percentiles respectively. **A.** Derived from F21, Figure S-4 (Syncrude-AU). **B.** Derived from F21, Figure S-4 (Shell-JP), **C.** Derived from F18, Figure 3 (Syncrude-ML). **D.** Derived from F17, Figure S-3 (CNRL).



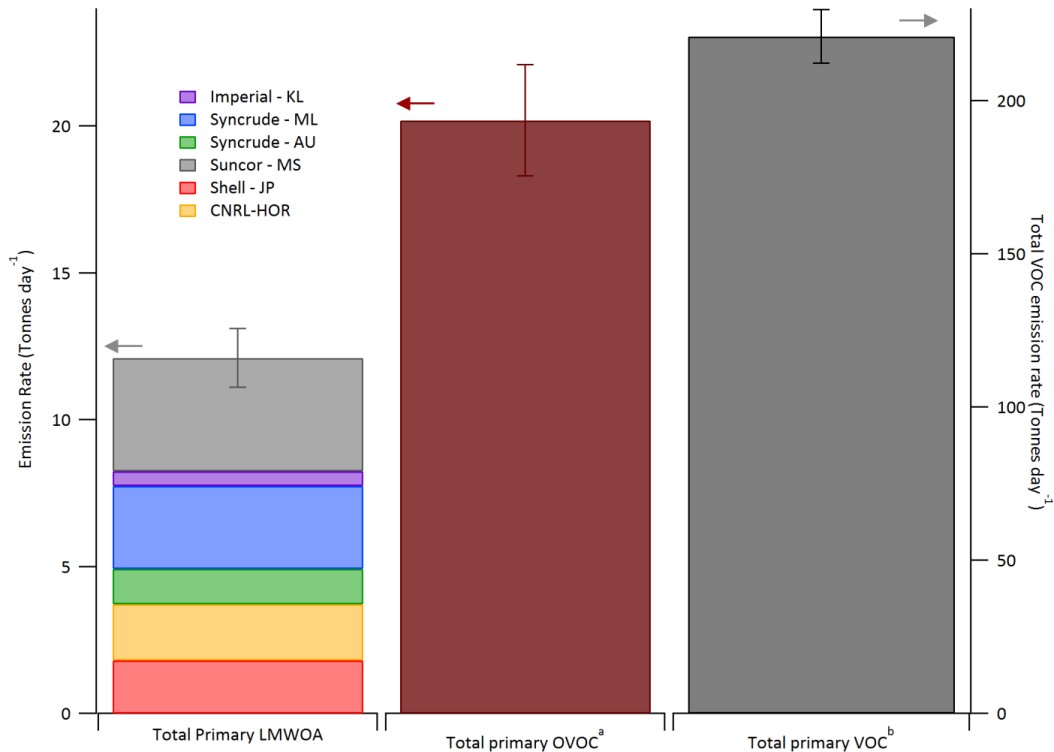

**Figure 5.** Total Daily primary LMWOA emitted in the OS (this study; left axis) compared to recently reported total primary VOC (right axis) and total primary oxygenated VOC emissions (left axis). **a**. As reported in Li et al., 2017. **b.** including oxygenates as reported in Li et al., 2017.



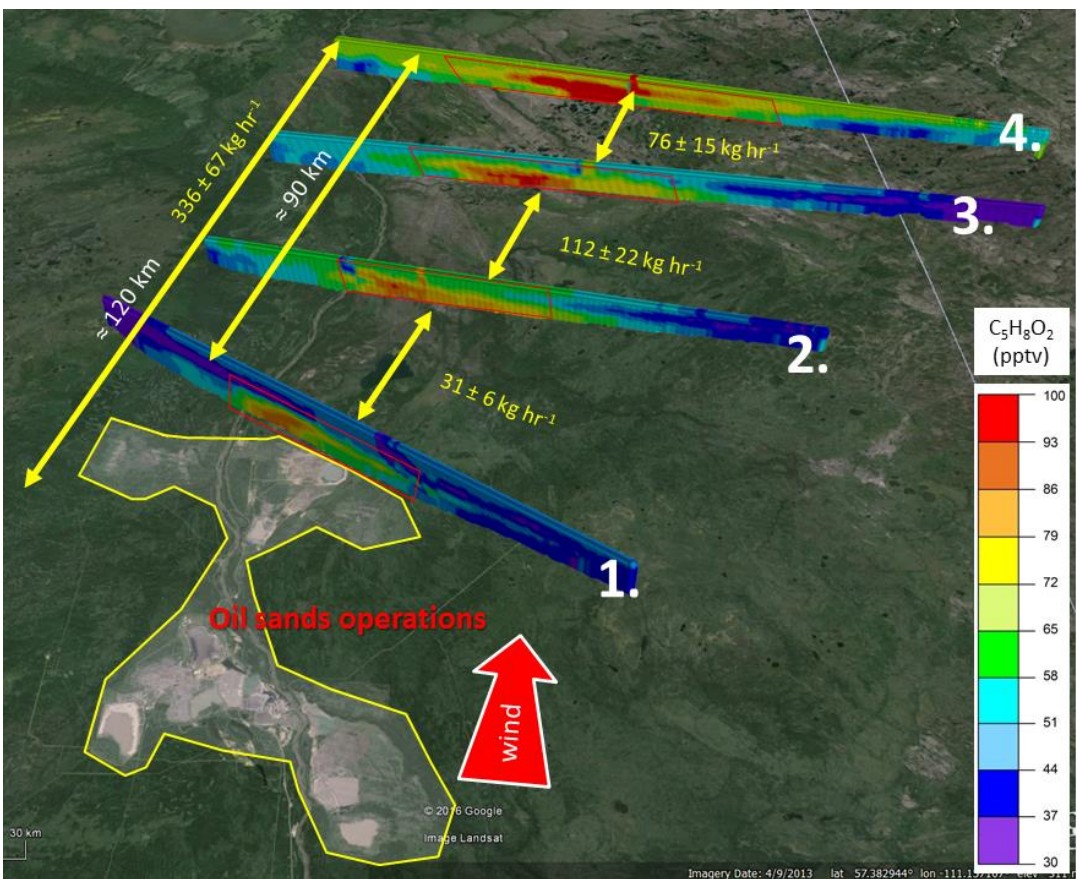

**Figure 6:** TERRA derived concentration screens for F19 using $C_5H_8O_2$ as a relevant LMWOA example. The LMWOA transfer rate difference between screens represents the secondary production rate of a given species (yellow text; 219±43 kg hr$^{-1}$ for screens 1 – 4). The overall rate from the OS source region is the integrated
5    LMWOA transfer rate through screen 4 after subtracting a small primary emission rate (336±67 kg hr$^{-1}$; see text).



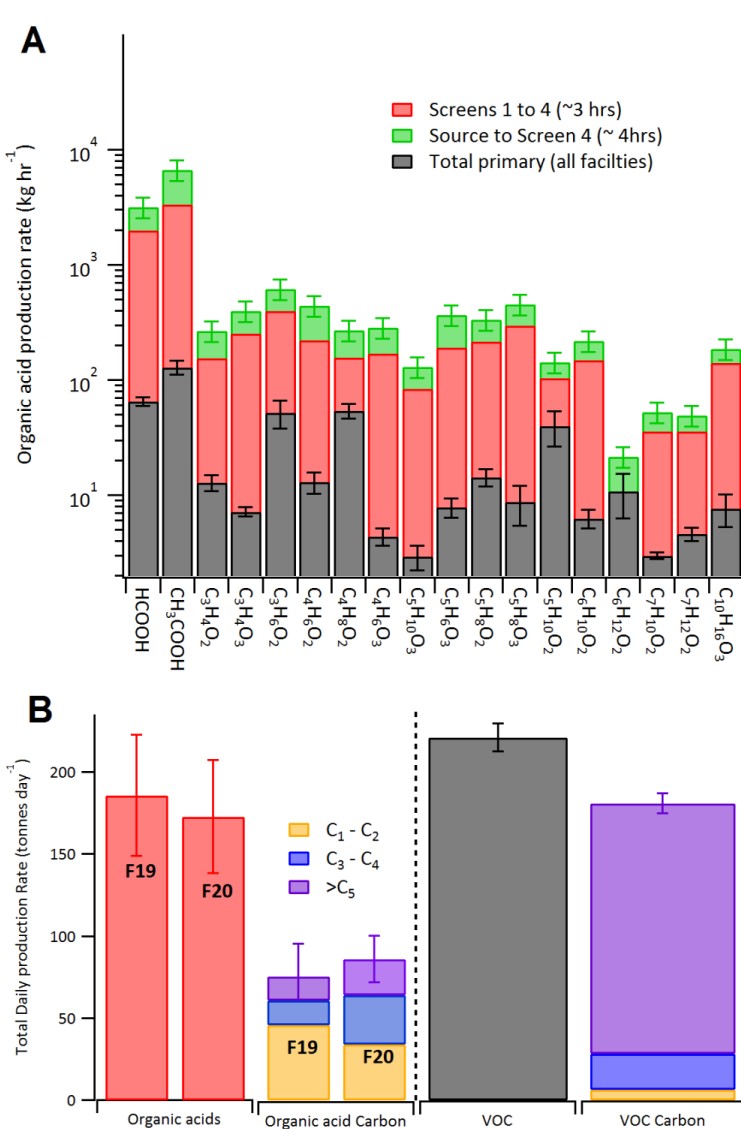

**Figure 7: A.** Secondary LMWOA production rates (kg hr$^{-1}$) derived between screens 1 and 4 of F19 using the TERRA algorithm, and at screen 4 after removing the primary LMWOA contribution (green bars). The primary LMWOA emission rates are also shown (Grey bars). **B.** Total LMWOA secondary production rates (F19 and F20 neglecting deposition) extrapolated to 1 photo-chemical day compared to total reported primary VOC emissions and carbon associated with these VOC emissions (S-M. Li et al., 2017).



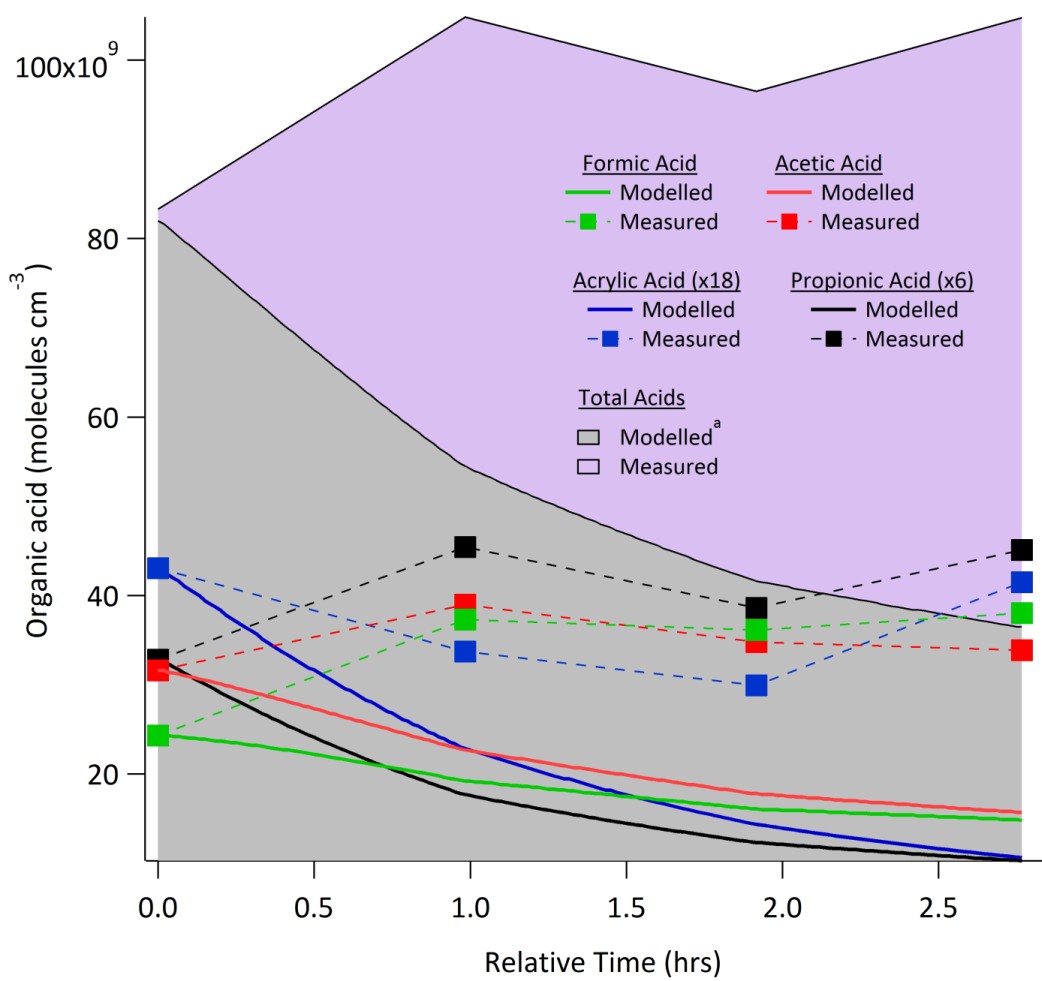

**Figure 8:** Comparison between specific measured and modelled organic acid species (Formic, acetic, acrylic and propionic) for F19. Total measured and modeled LMWOAs are shown as grey and purple shaded regions. **a.** Total modelled LMWOA concentration represents the sum of 27 species accounting for~ 99.8% of modelled LMWOA mass.

16
17





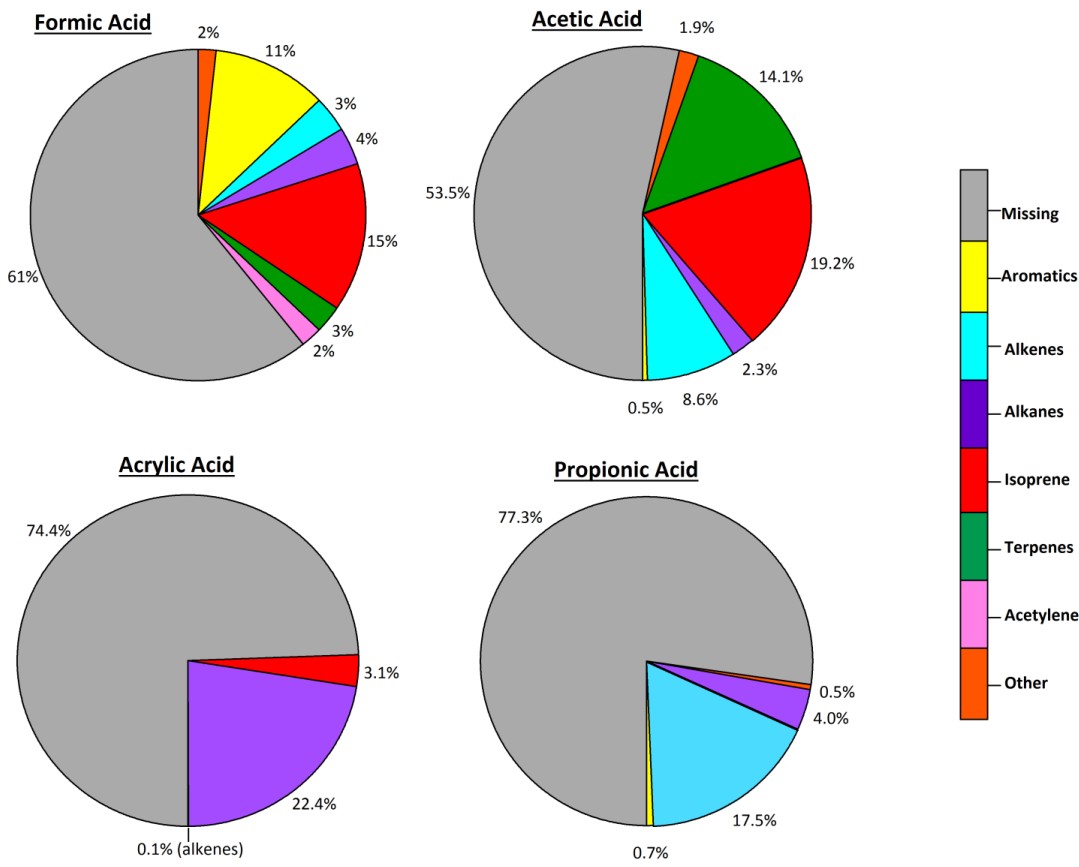

**Figure 9:** Relative contribution of various precursor hydrocarbon types to the modelled secondary production rate after 3 hours of evolution during F19, for formic, acetic, acrylic and propionic acids respectively.