# Peer review of "Understanding the Primary Emissions and Secondary Formation of Gaseous Organic Acids in the Oil Sands Region of Alberta, Canada"

_Atmospheric Chemistry and Physics, 2017_

## Referee Comment (RC1) · Anonymous Referee #1 · 12 Apr 2017

This papers presents the results of low molecular weight organic acids (LMWOA) measurements using an acetate ion time of flight mass spectrometer in an oil sands region near Alberta, Canada. The focus of the paper is to identify several organic acids observed downwind of the facilities and compare estimates of LMWOA emission and formation rates using observations from aircraft and compare the result to emission estimates derived using a box model. The results indicated that while oil sands regions are a large local source of secondary LMWOA, on a global source the estimated emissions account for <1% of the global secondary anthropogenic/biomass burning budget. A comparison to the box modelling results showed significant under prediction of secondarily formed LMWOA, which is in agreement with other previous reports of LMWOA

formation. The manuscript is well written and certainly of interest to the readership of Atmospheric Chemistry and Physics. It is a nice addition to the existing literature on LMWOA and also once again highlights the need for an improved understanding of the sources of these important secondary products. I support the publication of this manuscript in ACP subject to the following minor technical corrections.

**Comments:**

In the abstract the authors claim the emissions of the C1-C5 organic acids are directly from off-road diesel vehicles in the regions studied. However, at no time during the manuscript do the author make a case for the attribution to this versus other potential emission sources in the highly complex oil sands regions. They do in fact show that a primary source exists however, do not attribute that in a manner sufficient enough to make this claim. Do the authors have a comparison of observed BC/OA to previously published BC/OA ratios in diesel exhaust?

I may have missed this detail, but, why not directly calibrate for acetic acid using the LCU, instead opting to calculate the sensitivity using known kinetic rates?

Page2, line27, put a space before the citation

Page 4, line 22, the statement "quantify facility overall primary emissions" is awkward and needs editing

Page 6, line 19, suggest deleting the word 'with' in "derived with utilizing box-like"

Page 7, line 22, suggest to add some detail about the flight tracks being at varying altitudes to make that clear here.

A general figure map comment. In the figures where a red box is used to indicate the areas of mining activity it is difficult in some to see that this is a legend and not an indication of the area. One such example is figure 3. I think that Figure 1 does a good job of placing that legend so that it is not confusing. If possible I would recommend making all of the graph legends of the format and location on figure 1.

Figure 1. It would be nice to see a black trace of the flight tracks on this in altitude space so the user can visually see the degree of interpolation or smoothing of the model.

Figure 7. The bars in this case I believe are not stacked, though it is not explicitly mentioned. Could the authors clarify this in the figure description?

SI page 4, line 6, there is an incorrectly formatted alpha in alpha-pinene in my version.

СЗ

---

## Referee Comment (RC2) · Anonymous Referee #2 · 23 Apr 2017

This manuscript describes measurements of low molecular weight organic acids (LM-WOA) from the oil sands in northern Alberta, and uses a box model to evaluate the measurements. The estimates of emission ratios, and impacts of those emission ratios on concentrations downwind is a useful addition to the growing literature on organic acids in the atmosphere. I recommend publication following minor revisions.

Comments:

- There are too many acronyms to keep straight! I suggest moving most of them into full words (e.g. OS -> oil sands), and reserving acronyms for extremely long word combinations ('LMWOA') or very commonly used acronyms (VOC, SOA).

- The authors set up an excellent case for their study in the Introduction – a solid description of relevant literature, and lines of evidence regarding exactly why anthropogenic sources from oil production could be regionally and globally relevant. However, the last sentence of the introduction ("It is expected that the results of this study will be broadly applicable to the secondary formation of LMWOA from other anthropogenic activities with hydrocarbon emissions") does not follow this level of logic and reasoning. For example, other anthropogenic activities with hydrocarbon emissions can range from vehicle emissions to natural gas extraction to industrial solvent production to human-induced biomass burning activities, etc. The VOC precursor emissions from each of these sources can vary widely, and thus I expect the LMWOA emissions to vary as well. After explaining how unconventional the oil sands region is – and that the emissions from that region can be different from other areas – I don't see how these measurements will be broadly applicable to other regions. The authors have built up a strong case for their work, and broad applicability to other anthropogenic hydrocarbon sources is unnecessary to convince me that this work is important. I strongly recommend the authors delete the sentence.

- Authors identify the deposition of organic acids downwind of the oil sands as a potential source of atmospheric acidity. My recollection of the acid deposition literature is that organic acids do not contribute to ecosystem acidity – formic and acetic acids are weak acids, in contrast to nitric and sulphuric acids, which are strong acids. Can the authors point to literature or back-of-the-envelope estimates to further back up this idea that deposition of organic acids could be relevant for ecosystems downwind of the study area? While I understand that this is not the focus of the paper, and a detailed estimation is beyond the scope of the paper, the suggestion that organic acids could influence ecosystem acidity is a large one, and I think that the authors would do well to provide a little more literature or evidence that this is a likely event.

Technical Comments

- pg.13, line 6: misspelled Lagrangian.

- Figure 4, Box D has the note that the blue square indicates the mean value; it is distracting to have this in the middle of the yellow bar – I suggest either removing the legend from the figure and merely describing it in the figure caption, or offsetting it from the data columns so that it can not be confused with actual data

- Figure 9. Please specify that these results are from the photochemical box modeling, as opposed to observationally-derived for clarity

---

## Author Comment (AC1) · 19 May 2017

We appreciate the careful consideration of our manuscript by the reviewers. We have carefully responded to all of the point-by-point comments and issues raised by the reviewers and have revised the manuscript accordingly. These revisions are described in detail below, with our responses given in blue.

**Responses to Reviewer #1:**

This paper presents the results of low molecular weight organic acids (LMWOA) measurements using an acetate ion time of flight mass spectrometer in an oil sands region near Alberta, Canada. The focus of the paper is to identify several organic acids observed downwind of the facilities and compare estimates of LMWOA emission and formation rates using observations from aircraft and compare the result to emission estimates derived using a box model. The results indicated that while oil sands regions are a large local source of secondary LMWOA, as a global source the estimated emissions account for <1% of the global secondary anthropogenic/biomass burning budget.

A comparison to the box modelling results showed significant under prediction of secondarily formed LMWOA, which is in agreement with other previous reports of LMWOA formation. The manuscript is well written and certainly of interest to the readership of Atmospheric Chemistry and Physics. It is a nice addition to the existing literature on LMWOA and also once again highlights the need for an improved understanding of the sources of these important secondary products. I support the publication of this manuscript in ACP subject to the following minor technical corrections.

We thank the reviewer for his/her positive comments. We agree that this paper will make a good addition to existing literature on this topic. We have addressed the specific issues raised by this reviewer below.

In the abstract the authors claim the emissions of the C1-C5 organic acids are directly from off-road diesel vehicles in the regions studied. However, at no time during the manuscript do the author make a case for the attribution to this versus other potential emission sources in the highly complex oil sands regions. They do in fact show that a primary source exists however; do not attribute that in a manner sufficient enough to make this claim. Do the authors have a comparison of observed BC/OA to previously published BC/OA ratios in diesel exhaust?

While we agree that the OS facilities are highly complex, we believe that the majority of evidence we have presented demonstrates that a small portion of the LMWOA observed are indeed from off-road diesel. However, we do agree that we did not clearly make this obvious by perhaps synthesizing all the various forms of evidence into one clear statement in the manuscript.

There are several key points that suggest a primary source of off-road diesel. First, there is a relatively strong correlation with BC in the OS plumes, and we know that the vast majority of BC in the OS is from off-road diesel fuel use. The correlation with BC can be seen in Figure 2.

Secondly, the wind directions during the flights combined with the location of the measured LMWOA suggest a source directly from the mine where the off-road diesel is used. This can be seen in figure 1. Thirdly, flights directly over the mines (and not elsewhere) indicate that there are LMWOA emissions at that point, and further secondary formation of LMWOA moving away from the primary emission point. This can be seen in Figure 4. Finally, an LMWOA emission from off-road diesel is a known emission, and it should be expected to exist from the off-road diesel trucks used in the mines here as well. While we have not explicitly ruled out a co-emitting contribution from evaporating OS ore/bitumen in the mine itself (where the trucks are), this is highly unlikely as the oxygen content of bitumen is very small and evaporative emissions are dominantly hydrocarbon in nature. It may be possible that some LMWOA are formed in the mine heterogeneously (ie: from surface oxidation of ore) and misrepresented as being from the off-road diesel. However, this would be observed as a more broad area source over the entire mine rather than in more discreet areas of the mine (where the trucks likely were).

We have synthesized the above points into several lines which have been added to the primary LMWOA section of the manuscript. We think this now more clearly lays out the reasoning for suggesting an off-road diesel source. The manuscript now reads as (pg 8, lines 11-29):

*"In the case of LMWOA, there is a primary emission source attributed to the use of off-road heavy duty diesel vehicles within the open pit mines, as suggested by several observations described below. Firstly, the prevailing winds and positioning of the concentration enhancements at the box wall of each flight indicate that the open pit mines are the most likely source of the LMWOA emissions (for example in Figure 1). Within the mines, emissions of any kind can arise from the vehicles within the mine, or via volatilization from the mine face itself. Emissions of LMWOA via volatilization is highly unlikely as the mined oil sands material is extremely rich in hydrocarbons and deficient in oxygenates, and would nonetheless be observed as a more spatially homogenous source (which was not evident). The same is also true for the potential heterogeneous oxidation of the oil sands ore leading to LMWOA. Secondly, LMWOA during box flights were consistently correlated with black carbon (BC), and not correlated with other species such as $SO_2$. An example of this relationship is given in Figure 2-A for propionic acid during F18. Figure 2-A demonstrates that there is a high degree of similarity between the time series of propionic acid and BC, particularly at the plume intercepts on the virtual box wall. Since the majority of black carbon emissions within oil sands operations arise from the heavy hauler trucks used in the open pit mining (Wang et al., 2016), the observed LMWOA:BC correlation is consistent with an emission from the large diesel heavy hauler trucks within the mine and with the known emissions of various LMWOA from diesel fuel combustion (Kawamura et al., 2000;Wentzell et al., 2013;Crisp et al., 2014;Zervas et al., 2001b). Finally, flights directly over the mines indicate that there are LMWOA emissions at that point specifically (and not elsewhere), and further secondary formation of LMWOA moving away from the primary emission point (see section 3.1.1). "*

I may have missed this detail, but, why not directly calibrate for acetic acid using the LCU, instead opting to calculate the sensitivity using known kinetic rates?

We agree it would have been ideal to directly calibrate the PTR-MS for Acetic acid with the LCU. However, acetic acid was not on the original list of compounds to be measured and thus was not calibrated at the time of the study. Furthermore, post-calibration of the PTR-MS with the LCU was not possible as it was deployed elsewhere at the point in time when a measurement of acetic acid was known to be relevant for this paper. When acetic acid was finally realized to be important for this manuscript the PTR-MS had too many modifications/changes to make an LCU relevant to this study. We believe the relative approach we have taken should provide us with reasonably good acetic acid measurements, given that the response factor derived is within 10% of the RF determined with a permeation tube for a previous study under similar conditions (noted in the paper, pg 6 line 4).

Page2, line27, put a space before the citation

We have now added a space before the citation.

Page 4, line 22, the statement "quantify facility overall primary emissions" is awkward and needs editing

We agree, and have changed the wording of this sentence as follows:

 *"Thirteen of the flights were conducted to quantify the total primary emissions for each facility by flying in the shape of a four or five sided polygon, at multiple altitudes, resulting in 21 separate virtual boxes around 7 oil sands facilities."*

Page 6, line 19, Suggest deleting the word 'with' in "derived with utilizing box-like"

We agree and have removed the word "with".

Page 7, line 22, suggest adding some detail about the flight tracks being at varying altitudes to make that clear here.

The box model did not use the flight screen data at varying altitudes. This was a lagrangian box model such that the same point in the plume was sampled 1 hour apart. The description of how these observation points were determined is noted in the text of the same paragraph:

*"The specific data points within the plume to simulate with the model were selected to be those which were very close to being truly Lagrangian in nature as determined from the wind speed, wind direction and flying time. The successive plume intercepts modelled here are depicted in Figure S-1, where the same air parcel was sampled typically within 1 minute of its arrival time based on the wind."*

A general figure map comment. In the figures where a red box is used to indicate the areas of mining activity it is difficult in some to see that this is a legend and not an indication of the area. One such example is figure 3. I think that Figure 1 does a good job of placing that legend so that

it is not confusing. If possible I would recommend making all of the graph legends of the format and location on figure 1.

We agree that the graph legends could be better placed. We have now moved the legends in Figures 3 and S2 to be similar to Figure 1.

Figure 1. It would be nice to see a black trace of the flight tracks on this in altitude space so the user can visually see the degree of interpolation or smoothing of the model.

We agree and have added the suggested flight tracks.

Figure 7. The bars in this case I believe are not stacked, though it is not explicitly mentioned. Could the authors clarify this in the figure description?

The bars are indeed stacked and we have now clarified this in the figure caption.

SI page 4, line 6, there is an incorrectly formatted alpha in alpha-pinene in my version.

Thanks for catching that. We have fixed this error in the revised manuscript.

---

## Author Comment (AC2) · 19 May 2017

We appreciate the careful consideration of our manuscript by this reviewer. We have carefully responded to all of the point-by-point comments and issues raised by the reviewer and have revised the manuscript accordingly. These revisions are described in detail below, with our responses given in blue.

**Responses to Reviewer #2:**

This manuscript describes measurements of low molecular weight organic acids (LMWOA) from the oil sands in northern Alberta, and uses a box model to evaluate the measurements. The estimates of emission ratios, and impacts of those emission ratios on concentrations downwind is a useful addition to the growing literature on organic acids in the atmosphere. I recommend publication following minor revisions.

We thank the reviewer for his/her positive comments. We agree that this paper will make a good addition to existing literature on this topic. We have addressed the specific issues raised by this reviewer below.

There are too many acronyms to keep straight! I suggest moving most of them into full words (e.g. OS -> oil sands), and reserving acronyms for extremely long word combinations ('LMWOA') or very commonly used acronyms (VOC, SOA).

We agree that there are many acronyms in this paper. We have taken the reviewer's suggestion and tried to use full words where possible. Specifically "OS" is now "oil sands", PAN is now "peroxyacetyl nitrate", FA is now "formic acid", AA is now "acetic acid".

The authors set up an excellent case for their study in the Introduction – a solid description of relevant literature, and lines of evidence regarding exactly why anthropogenic sources from oil production could be regionally and globally relevant. However, the last sentence of the introduction ("It is expected that the results of this study will be broadly applicable to the secondary formation of LMWOA from other anthropogenic activities with hydrocarbon emissions") does not follow this level of logic and reasoning. For example, other anthropogenic activities with hydrocarbon emissions can range from vehicle emissions to natural gas extraction to industrial solvent production to human-induced biomass burning activities, etc. The VOC precursor emissions from each of these sources can vary widely, and thus I expect the LMWOA emissions to vary as well. After explaining how unconventional the oil sands region is – and that the emissions from that region can be different from other areas – I don't see how these measurements will be broadly applicable to other regions. The authors have built up a strong case for their work, and broad applicability to other anthropogenic hydrocarbon sources is unnecessary to convince me that this work is important. I strongly recommend the authors delete the sentence.

We agree that "broadly applicable" is an overstatement. While we agree on this point, we note that it is the underestimate of LMWOA formation from the box model that is likely to be more broadly applicable even to urban areas, since the VOCs oxidation mechanisms used here are

not going to be significantly different for any other location (even though the amounts and types of LMWOAs formed will be different). Regardless, for clarity we have removed this sentence from the paper since as the reviewer states, it is not necessary.

Authors identify the deposition of organic acids downwind of the oil sands as a potential source of atmospheric acidity. My recollection of the acid deposition literature is that organic acids do not contribute to ecosystem acidity – formic and acetic acids are weak acids, in contrast to nitric and sulphuric acids, which are strong acids. Can the authors point to literature or back-of-the-envelope estimates to further back up this idea that deposition of organic acids could be relevant for ecosystems downwind of the study area? While I understand that this is not the focus of the paper, and a detailed estimation is beyond the scope of the paper, the suggestion that organic acids could influence ecosystem acidity is a large one, and I think that the authors would do well to provide a little more literature or evidence that this is a likely event.

We agree that strong acidity will dominate ecosystem acidity in many cases, and we have indeed noted this point in the manuscript already (lines 26-27): *"The strong acidity associated with sulfur deposition is likely to dominate downwind ecosystem effects. Consequently, critical load exceedances for highly sensitive aquatic systems in Northern Alberta have mainly been assessed from the perspective of sulfur and nitrogen"*

However, weak acidity has been considered as a possible acidifying agent particularly for remote areas where strong acidity is mostly absent. We have now added literature references for the consideration of weak acids in acidification (although there are admittedly few of them). Downwind of the oil sands, most of the highly sensitive lakes/rivers etc are more likely to be impacted by inorganic acidity. However the degree of LMWOA impact depends upon the critical load that can be tolerated by the ecosystem. In the event that some lakes are just slightly below their critical load (assessed from the standpoint of S deposition), an additional input of even weak acidity may bump the ecosystem into a critical load exceedance. This would of course need to be verified and explored further, but is possible. We have clarified the text to make this point clearer. The paragraph now reads:

*"The strong acidity associated with sulfur deposition is likely to dominate downwind ecosystem effects. However, the impact of weak acidity on ecosystem acidification has been previously considered* (Vet et al., 2014)*, and may be particularly important in remote areas* (Stavrakou et al., 2012;Keene, 1983). *The impact of weak acidity downwind of the oil sands specifically is not clear. Critical load exceedances for highly sensitive aquatic systems in Northern Alberta have mainly been assessed from the perspective of sulfur and nitrogen (Cathcart et al., 2016;Whitfield et al., 2016). While the impact of the large amount of weak organic acidity formed downwind of the oil sands has not been evaluated, it may have a relevant impact on ecosystem acidification in highly sensitive systems which are approaching their respective sulfur and/or nitrogen critical loads. These results warrant a further investigation of the potential impact of this LMWOA emission/formation in this context"*

pg.13, line 6: misspelled Lagrangian.

We have fixed the spelling here.

Figure 4: Box D has the note that the blue square indicates the mean value; it is distracting to have this in the middle of the yellow bar – I suggest either removing the legend from the figure and merely describing it in the figure caption, or offsetting it from the data columns so that it cannot be confused with actual data

We have now removed the legend and simply described it in the figure caption.

Figure 9: Please specify that these results are from the photochemical box modeling, as opposed to observationally-derived for clarity

We have made the suggested change.

---

## Author Response (AR2)

Thank you for the quick response.

All the changes requested by the co-editor have been completed:

1. "significantly" to "sufficiently"

2. "OS" acronyms spelled out in supplemental information text

3. Space added after semi colon in all reference citations in text